# Targeted Therapy and Immunotherapy in Early-Stage Non-Small Cell Lung Cancer: Current Evidence and Ongoing Trials

**DOI:** 10.3390/ijms23137222

**Published:** 2022-06-29

**Authors:** Marco de Scordilli, Anna Michelotti, Elisa Bertoli, Elisa De Carlo, Alessandro Del Conte, Alessandra Bearz

**Affiliations:** 1Department of Medicine (DAME), University of Udine, 33100 Udine, Italy; marco.descordilli@cro.it (M.d.S.); anna.michelotti@cro.it (A.M.); elisa.bertoli@cro.it (E.B.); 2Department of Medical Oncology, Centro di Riferimento Oncologico di Aviano (CRO), IRCCS, 33081 Aviano, Italy; elisa.decarlo@cro.it (E.D.C.); alessandro.delconte@cro.it (A.D.C.)

**Keywords:** non-small cell lung cancer, early stage, adjuvant therapy, neoadjuvant therapy, targeted therapy, tyrosine kinase inhibitors, EGFR mutations, ALK rearrangements, immunotherapy, immune checkpoint inhibitors

## Abstract

The scenario of neoadjuvant and adjuvant settings in non-small cell lung cancer (NSCLC) is rapidly evolving. As already happened for the advanced disease, also early stages have entered the era of precision medicine, with molecular analysis and Programmed death-ligand 1 (PD-L1) evaluation that by now can be considered a routine assessment. New treatment options have been recently approved, with osimertinib now part of clinical practice for Epidermal Growth Factor Receptor mutated (EGFRm) patients, and immune checkpoint inhibitors (ICIs) available after FDA approval both in the adjuvant (atezolizumab) and neoadjuvant (nivolumab) setting. No mature data on overall survival benefits are available yet, though. Several clinical trials with specific-tyrosine kinase inhibitors (TKIs) and ICIs are currently ongoing, both with and without concomitant chemotherapy. As therapeutic strategies are rapidly expanding, quite a few questions remain unsettled, such as the optimal duration of adjuvant targeted therapy or the effective benefit of ICIs in early-stage EGFRm or ALK (Anaplastic Lymphoma Kinase) rearranged patients, or the possibility to individuate high-risk patients after surgical resection assessing minimal residual disease (MRD) by ctDNA evaluation. We hereby report already available literature data and summarize ongoing trials with targeted therapy and immunotherapy in early-stage NSCLC, focusing on practice-changing results and new perspectives for potentially cured patients.

## 1. Introduction

A platinum-based two-drug combination chemotherapy (CT) represents the standard adjuvant treatment in resected non-small cell lung cancer (NSCLC) with a pathologic stage II or III according to the American Joint Committee on Cancer (AJCC) tumor-nodes-metastases (TNM) classification. Survival benefit for cisplatin-combinations has been quantified at about 5.4% at 5 years by the Lung Adjuvant Cisplatin Evaluation (LACE) meta-analysis [1]. In another meta-analysis, Burdett et al. reported an absolute increase in survival at 5 years of about 4% [2]. Overall survival rates at 5 years range from 90% to 12% according to the stage at diagnosis (from IA to IIIC) [3].

CT has remained the only therapeutic standard-of-care option in the adjuvant setting for almost two decades. Only in most recent years, with the revolution of immunotherapy and new targeted therapies for oncogene-addicted disease in the advanced setting, the opportunity to exploit the possible benefit deriving from these new treatment options in earlier stages has been taken into consideration. In this way, while molecular testing and Programmed death-ligand 1 (PD-L1) evaluation have become mandatory over the years in advanced NSCLC to identify predictive factors for new therapies, it has not been historically required as a routine in stage I-III disease: it represents only a recent addiction in early-stage NSCLC patients (pts) due to the several clinical trials conducted both in the adjuvant and neoadjuvant setting.

Recent trials have been testing the efficacy both of driver mutation-specific tyrosine kinase inhibitors (TKIs) and immune checkpoint inhibitors (ICIs) (Figure 1). As already happened in the advanced setting, for oncogene-addicted disease, experimentations have focused on Epidermal Growth Factor Receptor (EGFR) gene mutations (EGFRm) and Anaplastic Lymphoma Kinase (ALK) rearrangements (ALKr), the most commonly detected in NSCLC pts. It has been reported that the prevalence of EGFRm is mostly preserved throughout disease history, with a similar prevalence in early and advanced diseases [4]. On the contrary, ALKr tend to be reported as less frequent in earlier stages [5,6], probably as a consequence of a more aggressive clinical behavior with rapidly developing metastases. Therefore, EGFRm are expected in up to 15% of early-stage NSCLC Caucasian pts, while ALKr can be found in less than 5% of them.

Far less common are ROS Proto-Oncogene 1 (ROS1), RET or Neurotrophic Tyrosine Receptor Kinase (NTRK) 1–3 genes rearrangements, or BRAF, HER2 or MET gene alterations. All these drivers, as well as the more common KRAS gene mutations, have not been a specific focus of research yet for phase III clinical trials in early-stage NSCLC.

As regards immunotherapy trials, pharmacological agents have been variably tested both considering and irrespective of PD-L1 evaluation on tumor cells.

In the numerous completed or still ongoing trials with TKIs and ICIs, drugs have been experimented with after standard platinum-based adjuvant CT or as a replacement to it. This still remains an open question, whether postoperative CT can be actually superseded in biomarker-selected pts candidates to adjuvant new generation therapies.

As of today, there is no validated method to more accurately identify pts at high risk of disease relapse after surgery other than classical TNM pathologic staging. Various techniques are currently being studied. As an example, several clinical trials have been considering the possibility to identify minimal residual disease (MRD) after radical resection, for instance by ctDNA evaluation. The presence of MRD is associated with reduced disease-free survival (DFS) [7]. However, all currently explored methods are still experimental and suffer from high costs, limited sensitivity, and/or little potential availability in clinical practice outside of hub centers or research facilities [8]. In our review, we will also report data on MRD-selected pts from completed or ongoing trials with TKIs or ICIs.

Finally, another open question is related to the role of drugs from these pharmacological classes in the neoadjuvant setting. Considering their usually high response rate in advanced stages, are they capable to improve tumor regression as a preoperative treatment, in comparison to CT alone? And will this eventually correlate with better survival outcomes?

The objective of our review is to report already available literature data and to summarize ongoing trials with targeted agents and immunotherapy in early-stage NSCLC, focusing on practice-changing results and new perspectives for potentially curable pts.

## 2. EGFR Tyrosine Kinase Inhibitors

### 2.1. Adjuvant Setting

It is reported that pts with common EGFR mutations (exon 19 deletion (Ex19del) and exon 21 L858R mutation) typically have a shorter DFS after radical surgery, even if receiving standard platinum-based adjuvant CT [9]. Also considering the poorer prognosis of these pts, several clinical trials have explored the DFS benefit of EGFR-TKIs in the adjuvant setting.

The BR19 trial (NCT00049543) was the first phase III trial to evaluate **gefitinib**, a first-generation EGFR-TKI [10]. 503 pts with stage IB-IIIA resected NSCLC and not selected by EGFRm received either oral gefitinib 250 mg die or placebo for up to 2 years after postoperative radiotherapy and eventual CT. No benefit was reported, neither in DFS (HR 1.22, 95% CI 0.93–1.61, *p* = 0.15), nor in overall survival (OS) (HR 1.24, 95% CI 0.94–1.64, *p* = 0.14). No benefit was evidenced also in the small subgroup of EGFRm pts (4 out of 359 with known EGFR status). The trial was prematurely closed (an enrollment of 1242 pts had been planned).

The phase III ADJUVANT-CTONG1104 trial (NCT01405079) enrolled 222 pts with resected stage II-IIIA EGFRm NSCLC [11]. They were randomized to gefitinib 250 mg die for 2 years or standard adjuvant CT with cisplatin-vinorelbine for 4 cycles. Median DFS was significantly longer in the experimental arm (30.8 vs. 19.8 months (m), HR 0.56, 95% CI 0.40–0.79, *p* = 0.001), with a DFS rate at 3 and 5 years of 39.6% vs. 32.5% and 22.6% vs. 23.2%, respectively. The benefit in DFS did not translate to survival, with a not statistically significant OS advantage (median of 75.5 vs. 62.8 m, HR 0.92, 95% CI 0.62–1.36, *p* = 0.674). Subsequent treatments received after disease relapse mostly contributed to OS (median not reached with other treatments received vs. 62.8 m with no other lines), especially if subsequent EGFR-TKIs were used (HR 0.23).

The phase III IMPACT trial (UMIN000006252) randomized 234 pts with stage II-III EGFRm NSCLC to 2 years of gefitinib or standard adjuvant CT [12]. The experimental arm showed a numerical benefit in DFS which was not statistically significant (median of 35.9 vs. 25.1 m, HR 0.92, 95% CI 0.67–1.28, *p* = 0.63); DFS rates at 5 years were 31.8% and 34.1% in the two arms. No difference in OS was reported (HR 1.03, 95% CI 0.65–1.65, *p* = 0.89; OS rates at 5 years 78.0% vs. 74.6%).

RADIANT phase III trial (NCT00373425) evaluated the benefit from another first-generation EGFR-TKI, **erlotinib** [13]. 973 stage IB-IIIA pts with EGFR-expressing tumors (either ≥ 1% staining at immunohistochemistry (IHC) or gene amplification at fluorescence in situ hybridization (FISH)) were randomized to erlotinib 150 mg die or placebo for 2 years after adjuvant CT. No significant difference in DFS (median of 50.5 vs. 48.2 m, HR 0.90, 95% CI 0.74–1.10, *p* = 0.324) and OS (median not reached, HR 1.13, 95% CI 0.88–1.45, *p* = 0.335) was reported between treatment arms. 161 pts (16.5%) were EGFRm and a DFS benefit was observed in this subgroup (median of 46.4 vs. 28.5 m, HR 0.61, 95% CI 0.38–0.98, *p* = 0.039), even if not statistically significant due to the hierarchical structure of the trial. Also, the phase II trial SELECT (NCT00567359) experimented erlotinib in stage IA-IIIA NSCLC pts [14]. 100 pts with EGFRm received erlotinib for 2 years, after adjuvant CT and eventual radiotherapy. Median DFS and OS were not reached, with 5-year DFS and OS rates of 56% (95% CI 45–66) and 86% (95% CI 77–92), respectively. The primary endpoint was a 10% improvement of the 2-year DFS rate in comparison to historical control, which was reached (88% vs. 76%, *p* = 0.0047).

EVAN (NCT01683175, phase II) assessed erlotinib in resected EGFRm NSCLC pts with stage IIIA only [15]. 102 pts were randomized to erlotinib for 2 years or standard adjuvant CT, with a reported benefit in DFS rate at 2 years of 36.7% (95% CI 15.5–58.0, *p* = 0.0007), 81.4% vs. 44.6% in the two arms, respectively (RR 1.82, 95% CI 1.19–2.78, *p* = 0.0054); median DFS was 42.4 vs. 21.0 m (HR 0.27, 95% CI 0.14–0.53, *p* < 0.0001). At a following update [16], a benefit was described also for OS (median of 84.2 vs. 61.1 m, HR 0.32, 95% CI 0.15–0.67), with a 5-year OS rate of 84.8% vs. 51.1%.

The possible benefit derived from **icotinib**, another EGFR-TKI, was evaluated in the EVIDENCE trial (NCT02448797, phase III), in which 322 stage II-IIIA EGFRm NSCLC pts were randomized to icotinib 125 mg × 3 die for 2 years or standard CT [17]. A DFS benefit was evidenced, with a median of 47.0 vs. 22.1 m (HR 0.36, 95% CI 0.24–0.55, *p* < 0.0001) and a 3-year DFS rate of 63.9% vs. 32.5%. Data on OS are still immature (HR 0.91, 95% CI 0.42–1.94).

Finally, ADAURA (NCT02511106) is a phase III trial in which 682 stage IB-IIIA EGFRm NSCLC pts were randomized to receive either **osimertinib** 80 mg die orally or placebo for up to 3 years, after having received or not adjuvant CT [18,19]. Considering stage II-IIIA pts only (470, i.e., 69%), in which DFS benefit was evaluated as the primary endpoint, median DFS was not reached but an 83% risk reduction for disease relapse or death was evidenced (HR 0.17, 99% CI 0.11–0.26, *p* < 0.001) at an interim analysis. DFS benefit was maintained also in the overall population (HR 0.20, 99% CI 0.14–0.30, *p* < 0.001) and was independent of disease stage and from having received adjuvant CT (60% of pts) or not (HR 0.16 with 95% CI 0.10–0.26 and HR 0.23 with 95% CI 0.13–0.40, respectively). OS data were immature, with a 2-year OS rate of 98% vs. 85%. A significant reduction in central nervous system (CNS) recurrence was reported in the experimental arm (HR 0.18, 95% CI 0.10–0.33).

In all the aforementioned trials enrolling EGFRm pts, common mutations (Ex19del and exon 21 L858R mutation) were considered. Data from all the completed/concluding clinical trials with EGFR-TKIs in the adjuvant setting are summarized in Table 1.

Several clinical trials with EGFR-TKIs in the adjuvant setting are currently ongoing (Table 2), evaluating the survival benefits both from first-, second-, or third-generation TKIs (i.e., gefitinib, erlotinib, icotinib, afatinib, osimertinib) and new molecules (i.e., furmonertinib, almonertinib).

Considering the available literature data from trials with concluded enrollment and completed or interim analyses, different EGFR-TKIs have shown a DFS benefit in EGFRm pts in comparison to standard adjuvant CT or as an addition to CT alone: gefitinib for 2 years after CT (ADJUVANT-CTONG1104), erlotinib for 2 years both after (SELECT, even if phase II and with immature data) or instead of CT (EVAN, phase II), icotinib for 2 years instead of CT (EVIDENCE), osimertinib for 3 years possibly after CT (ADAURA). However, no phase III trial has shown an OS benefit yet, with particular reference to all the trials with gefitinib, which already have mature data.

Several discussions have been made on the effective capacity of EGFR-TKIs to actually prevent disease recurrence, rather than simply delaying it. Keeping the focus on gefitinib, the DFS benefit appeared as a minimum, with disease relapse within 1 median year from experimental therapy completion; treatment arms had Kaplan-Meier survival curves crossing around 4 years after surgery and similar 5-year DFS rates (31.8% vs. 34.1% in IMPACT, 22.6% vs. 23.2% in ADJUVANT-CTONG1104). From these considerations, also a question on which is the optimal EGFR-TKI treatment duration arises, examining the brief disease-free interval after therapy interruption. ADAURA has been the only trial in which the adjuvant targeted therapy has been prolonged from 2 to 3 years.

Distinct meta-analyses have analyzed the comprehensive data and survival benefits with the different EGFR-TKis. Yin et al. [21] considered 11 studies with a total of 1900 EGFRm pts included and reported a DFS benefit with an HR of 0.42 (95% CI 0.31–0.57) and an OS benefit with an HR of 0.62 (95% CI 0.45–0.86). Chen et al. [22], with 7 randomized clinical trials considered and 1283 EGFRm pts included, reported similar data for DFS benefit (HR 0.41, 95% CI 0.24–0.70, *p* = 0.001) but no statistically significant OS benefit (HR 0.72, 95% CI 0.37–1.41, *p* = 0.336).

The published data with osimertinib certainly appear encouraging, with an unprecedented benefit in terms of DFS: an HR of 0.17 in the population considered for the primary endpoint and of 0.20 in the overall population. ADAURA also presented an important reduction in CNS recurrence in comparison to other EGFR-TKIs, which is coherent with the known superior CNS activity of osimertinib [23]. In this way, considering efficacy and safety data, ADAURA brought to a change in clinical practice, with the Food and Drug Administration (FDA) and European Medicines Agency (EMA) approval of osimertinib as the first targeted therapy available in the adjuvant setting for NSCLC pts. Longer follow-up and OS data maturity are anyway expected to confirm the magnitude of its effectiveness.

Since ADAURA allowed osimertinib administration both with and without previous adjuvant CT (and DFS benefit appeared as independent from CT) and considering that other trials showing DFS benefit (ADJUVANT-CTONG1104, EVAN, EVIDENCE) used EGFR-TKIs without CT, there is still no conclusive answer to whether these drugs should be employed with or without other antineoplastic agents.

### 2.2. Neoadjuvant Setting

The already available published data in the neoadjuvant setting derive from small phase II trials with first-generation EGFR-TKIs. NCT00188617 [24] was the first one to evaluate neoadjuvant **gefitinib** for 28 days in 36 stage I NSCLC pts non selected for EGFRm. In NCT00600587 [25], stage IIIA(N2) EGFRm pts were assigned to neoadjuvant **erlotinib** while pts without EGFRm received only CT. Objective response rate (ORR) was numerically higher in the experimental arm (58.3% vs. 25.0%). Also NCT01217619 [26] evaluated neoadjuvant erlotinib in the same setting of stage IIIA(N2) EGFRm pts, with a reported ORR of 42.1%.

In the EMERGING-CTONG1103 (NCT01407822) phase II trial [27], neoadjuvant erlotinib was compared with carboplatin-gemcitabine CT in stage IIIA EGFRm pts, with the possibility to continue the same therapy in each treatment arm also in the adjuvant setting. The 3-year and 5-year OS rates were 58.6% vs. 55.9% (*p* = 0.819) and 40.8% vs. 27.6% (*p* = 0.252), respectively.

Other trials are assessing the efficacy of targeted therapy started in the neoadjuvant setting and possibly continued even after surgery (Table 3). In particular, NeoADAURA (NCT04351555) is evaluating the benefit from neoadjuvant **osimertinib**, both in combination with CT for 3 cycles and alone for at least 9 weeks, in comparison to standard CT. CT and/or osimertinib can then be considered also in the adjuvant setting.

## 3. ALK Tyrosine Kinase Inhibitors

As described for EGFRm pts, it has been reported that also ALKr are associated with a worse prognosis in resected NSCLC pts [29,30]. However, literature data are not univocal and even if ALKr tumors are described as clinically aggressive, often with lymph nodes involvement despite low T stage [31], the effective prognostic significance of ALKr in resected NSCLC remains unsettled [5,32,33].

In comparison to the EGFRm disease, far less clinical trials have been investigating the role of specific TKIs in the ALKr early disease, with no currently published data available. Several trials are presently ongoing (Table 4).

ALCHEMIST (NCT02194738) is a large phase III trial in which 8300 pts with resected stage IB-IIIA NSCLC are tested for gene driver alterations in order to optimize their adjuvant treatment with targeted therapies [34]. Pts in the ALKr arm are being randomized to **crizotinib** 250 mg × 2 die for 2 years or observation after standard adjuvant CT.

Phase III ALINA trial (NCT03456076) is randomizing stage IB-IIIA ALKr NSCLC pts to either **alectinib** 600 mg × 2 die for 2 years or adjuvant CT [35].

Other clinical trials are evaluating the benefit of **ensartinib**, a second-generation ALK-TKI, in resected NSCLC pts with a scheme of 225 mg die, both as an alternative to standard adjuvant CT (NCT05341583, phase III trial with ensartinib for 2 years versus placebo; NCT05186506, phase II trial with ensartinib for 2 years versus adjuvant CT) or after it (NCT05241028, phase II trial with ensartinib for 3 years).

Finally, other trials are studying ALK-TKIs in ALKr pts even before surgery. In the ALNEO phase II trial (NCT05015010), stage III NSCLC pts are being administered alectinib 600 mg × 2 die for 8 weeks before and 96 weeks after surgery, for a total of 2 years [36]. NAUTIKA1 (NCT04302025) is a phase II trial in which stage IB-III NSCLC pts are biomarker-selected and receive a neoadjuvant and adjuvant therapy according to their oncogene driver alteration. In the ALKr cohort, pts are receiving alectinib for up to 8 weeks before surgery, then an adjuvant treatment with standard CT and alectinib for up to 2 years. Other cohorts include pts with ROS1, NTRK, RET alterations or BRAF mutations, receiving with the same scheme entrectinib, pralsetinib or vemurafenib + cobimetinib, respectively. The LIBRETTO-432 phase III trial (NCT04819100) is evaluating selpercatinib in stage IB-IIIA pts with RET fusion-positive NSCLC.

## 4. Immune Checkpoint Inhibitors

As regards immune checkpoint inhibitors, several trials have been designed and are currently ongoing to investigate their role in early-stage NSCLC.

### 4.1. Adjuvant Setting

The first positive results from a phase III trial in the adjuvant setting have been provided by IMpower010 (NCT02486718), in which 1280 stage IB-IIIA pts were randomized to 1 year of **atezolizumab** (1200 mg q21 for 16 cycles) or observation after standard cisplatin-based adjuvant CT [37]. Pts were enrolled independently from histology, EGFRm or ALKr and PD-L1 status. A DFS benefit was evidenced in the experimental arm of stage II-IIIA pts with PD-L1 ≥ 1% (HR 0.66, 95% CI 0.50–0.88, *p* = 0.0039), which was the primary endpoint (PD-L1 expression defined by SP263 assay). The DFS benefit was confirmed in the stage II-IIIA population independently from PD-L1 expression (HR 0.79, 95% CI 0.64–0.96, *p* = 0.02) and in the intention-to-treat population (HR 0.81, 95% CI 0.67–0.99, *p* = 0.04). It was even higher in pts with PD-L1 ≥ 50% (HR 0.43, 95% CI 0.27–0.68). No clear benefit was evidenced in the EGFRm and ALKr subgroups. OS data are still immature and OS was not formally tested due to the hierarchic design of the study, though preliminary stratified HR was 0.77 (95% CI 0.51–1.17). A longer follow-up is needed to confirm the translation of DFS benefit on survival. According to the positive reported results, atezolizumab received the FDA approval for the adjuvant treatment of PD-L1 ≥ 1% stage II-IIIA NSCLC pts. A positive opinion has also been adopted by EMA to support atezolizumab approval for high-risk NSCLC pts with PD-L1 ≥ 50% not harboring EGFRm or ALKr.

PEARLS/KEYNOTE-091 (NCT02504372) is another key phase III trial in the adjuvant setting in which **pembrolizumab** 200 mg q21 for 18 cycles has been tested against placebo in 1177 stage IB-IIIA NSCLC pts, after standard CT [38]. Also in this case, results from an interim analysis have shown a DFS benefit in the experimental arm, with a median of 53.6 vs. 42.0 m (HR 0.76, 95% CI 0.63–0.91, *p* = 0.0014). Surprisingly, the DFS benefit was not confirmed in the PD-L1 ≥ 50% (tumor proportion score, TPS) population (HR 0.82; 95% CI 0.57–1.18, *p* = 0.14). OS data are still immature, with a non-significant trend in favor of the experimental arm (18-month rate 91.7% vs. 91.3%, HR 0.87, *p* = 0.17). Based on these preliminary data (full publication is not available yet), pembrolizumab appears as another feasible option in this setting, independently from PD-L1 expression.

Clinical trials with already available data with ICIs in the adjuvant (and neoadjuvant) setting are summarized in Table 5.

Several other trials are still ongoing (see Table 6).

NCT04367311 is a phase II trial evaluating atezolizumab in addition to standard CT in stage IB-IIIA pts with detectable ctDNA after surgery. The primary endpoint is the percentage of pts with ctDNA negativization after treatment completion.

The ACCIO phase III trial (NCT04267848) is characterized by a design similar to PEARLS/KEYNOTE-091, with pembrolizumab evaluated in the adjuvant setting both sequentially after standard CT or started concomitantly with CT, for a total of 17 cycles [41]. Another phase II trial (NCT04317534) is studying pembrolizumab as adjuvant treatment in stage I NSCLC pts.

Other trials are evaluating **nivolumab** in this setting. ANVIL (NCT02595944) is a large phase III trial with a design similar to IMpower010 and PEARLS/KEYNOTE-091, with 1 year of nivolumab compared to observation after standard CT [42]. NADIM-ADJUVANT (NCT04564157, phase III) is assessing the same drug started concomitantly with CT [43].

BR31 phase III trial (NCT02273375) is randomizing stage IB-IIIB(N2) pts to either **durvalumab** or placebo for 1 year, taking as the primary endpoint DFS in pts with PD-L1 ≥ 25% expressing tumors. The same ICI is being evaluated in MERMAID-1 (NCT04385368, phase III), started together with adjuvant CT and compared to placebo [44]. In this case, the primary endpoint will be DFS in the subgroup of pts with detectable ctDNA after surgery. Also, MERMAID-2 (NCT04642469, phase III) is enrolling only pts with these characteristics, after possible neoadjuvant and/or adjuvant CT, randomizing them to 2 years of durvalumab or placebo [45].

Finally, other molecules are being studied, such as **toripalimab**, a new anti-PD-1 which is being tested against placebo in the phase III trial LungMate-008 (NCT04772287) after adjuvant CT. Other immune-modulating molecules are being evaluated. For example, CANOPY-A (NCT03447769) is a phase III trial assessing **canakinumab**, an anti-IL-1β antibody, after adjuvant CT in stage IIIA-IIIB pts.

### 4.2. Neodjuvant Setting

Important data with ICIs are already available also in the neoadjuvant setting (Table 5).

LCMC3 (NCT02927301) was the first study to present positive results, even if as a phase II trial [39]. 181 stage IB-IIIB(N2) pts were enrolled and administered neoadjuvant **atezolizumab** 1200 mg q21 for 2 cycles, then for another year after surgery. Survival data are not available yet, though major complete response was registered in 21% of pts without EGFRm or ALKr (primary endpoint), with 7% of them reaching pathologic complete response (pCR); R0 resection was performed in 92% of cases.

Recently, first results from CheckMate 816 (NCT02998528, phase III) were published [40]. 773 pts with stage IB-IIIA resectable NSCLC have been randomized to either neoadjuvant **nivolumab** 360 mg or placebo together with q21 platinum-based CT for 3 cycles. A third arm was initially planned with pts receiving nivolumab 3 mg/kg q14 for 3 cycles together with ipilimumab 1 mg/kg for 1 cycle, but it was prematurely closed. EGFRm and ALKr pts were excluded. CT for 4 cycles and/or radiotherapy was allowed after surgery. Event-free survival (EFS) was significantly longer in the experimental arm, with a median of 31.6 vs. 20.8 m (HR 0.63, 97.38% CI 0.43–0.91, *p* = 0.005); benefit was maintained even considering possible adjuvant therapy (HR 0.66, 95% CI 0.47–0.90). 24.0% vs. 2.2% of pts reached pCR in the two treatment arms (OR 13.94, 99% CI 3.49–55.75, *p* < 0.001), with a major pathologic response in 36.9% vs. 8.9% (OR 5.70, 95% CI 3.16–10.26). OS data are still immature, but a benefit trend was evidenced in the nivolumab arm at the first interim analysis (HR 0.57, 99.67% CI 0.30–1.07, *p* not statistically significant). A benefit was evidenced independently from PD-L1 expression, with a greater advantage in EFS in PD-L1 ≥ 1% pts. As an exploratory analysis, ctDNA levels were evaluated in 89 pts and its clearance was higher in the experimental arm (56% vs. 35%). Longer follow-up is needed but CheckMate 816 data certainly appear as practice-changing results in this setting and nivolumab combined with CT has already received FDA approval and is undergoing EMA centralized review procedure.

Many other trials are ongoing (Table 6).

IMpower-030 (NCT03456063) is a phase III trial randomizing stage II-IIIB(N2) pts to neoadjuvant atezolizumab/placebo in association with CT for 4 cycles; then, after unblinding, pts in the experimental arm will also receive adjuvant atezolizumab for 16 cycles [46]. Atezolizumab is being tested in this setting also in association with tiragolumab (an anti-TIGIT antibody) in the phase II NCT04832854 trial. Stage II-IIIB(N2) pts will receive the two antibodies and CT for 4 cycles, then the two experimental drugs will be continued for 16 more cycles in the adjuvant setting. In the biomarker-driven NAUTIKA1 trial (NCT04302025), 4 cycles of neoadjuvant atezolizumab are administered in pts without driver alterations and PD-L1 expression ≥ 1%.

In the KEYNOTE-671 phase III trial (NCT03425643), **pembrolizumab** is being compared to placebo in association with concomitant CT for 4 cycles, with then 13 more cycles of adjuvant pembrolizumab/placebo [47]. Pembrolizumab is being evaluated also without CT and together with lenvatinib for 6 weeks before surgery in the INNWOP1 phase II trial (NCT04875585), and then continued in the adjuvant setting for 15 cycles.

CheckMate 77T (NCT04025879) is studying the association of nivolumab with neoadjuvant CT but, unlike CheckMate 816, pts will continue nivolumab/placebo also in the adjuvant setting [48].

Finally, other trials are experimenting neoadjuvant **durvalumab**: in the NeoCOAST 2 (NCT05061550), in association with CT and either monalizumab (an ICI targeting Natural Killer Group 2A) or oleclumab (an anti-CD73 antibody); in the AEGEAN (NCT03800134), with concomitant CT and then continued as adjuvant therapy [49].

## 5. Discussion

The scenario of neoadjuvant and adjuvant setting in NSCLC is rapidly evolving. As already happened for the advanced disease, also early stages have entered the era of precision medicine, with molecular analysis and PD-L1 evaluation that by now can be considered part of the routine assessment. New treatment options are available, with adjuvant osimertinib which is now part of clinical practice for EGFRm pts (with FDA and EMA approvals), and the recent FDA approvals of atezolizumab after adjuvant CT for stage II-IIIA pts with PD-L1 ≥ 1%, and nivolumab in combination with neoadjuvant CT independently from PD-L1 expression. EMA approval is pending both for adjuvant atezolizumab in high-risk NSCLC pts with PD-L1 ≥ 50% and for neoadjuvant nivolumab in combination with CT. Pembrolizumab after standard CT is probably the next drug to be approved in the adjuvant setting, considering the promising data from PEARLS.

Possibly, the future challenge will be trying to incorporate the evaluation of other rare targetable alterations in the routine assessment of early-stage tumors, even considering that some of them could be quite rare in this setting, with inherent difficulties also in designing specific clinical trials.

Despite all these major innovations relative to new pharmacological classes, platinum-based CT still represents the only treatment option with confirmed benefit on overall survival in resected NSCLC pts. In fact, among all the available data from clinical trials in this setting, no benefit in overall survival in phase III trials has been reported yet. Considering the already evidenced benefit in DFS, data appear certainly promising for several drugs, with an expected OS benefit at data maturity, as testified by the aforementioned early FDA approvals. This is the case with osimertinib (ADAURA) in EGFRm pts, and with atezolizumab (IMpower010) and pembrolizumab (PEARLS/KEYNOTE-091) considering ICIs (Figure 2).

Regarding EGFR-TKIs, osimertinib definitely looks like the way-to-go, with a not ignorable 83% risk reduction in disease relapse or death and considering also the high proven activity in reducing CNS relapse. Controversies with EGFR-TKIs in this setting still regard the optimal treatment duration and the association with CT. Considering the early disease relapse after therapy completion reported in several trials with EGFR-TKIs, a longer treatment duration could possibly affect DFS benefit. ADAURA was the only trial extending the experimental treatment to 3 years, and even if available data already appear statistically solid, a longer follow-up is necessary to confirm the drug benefit in survival. No definitive answer is available also regarding EGFR-TKIs usage after or instead of adjuvant CT, considering that, while in ADAURA about 60% of pts received osimertinib after adjuvant CT, in other trials a DFS benefit was evidenced also without CT (ADJUVANT-CTONG1104, EVAN, EVIDENCE).

As regards ICIs, both atezolizumab and pembrolizumab are valid options after standard adjuvant CT. Atezolizumab already received FDA approval for pts with PD-L1 ≥ 1%, but a DFS benefit was evidenced independently from PD-L1 expression and was even higher in PD-L1 ≥ 50% pts. Also, the benefit from pembrolizumab appears independent from PD-L1 expression, even if, unexpectedly, DFS benefit was not confirmed in PD-L1 ≥ 50% pts. Full publications are needed, but as of now, there is no element making one drug preferable to the other. Also, considering the number of ongoing clinical trials, in the next future other ICIs are probably joining as possible therapeutic options in this setting, such as nivolumab (ANVIL) or durvalumab (BR31). To make things even more complex in treatment selection, ICIs have recently entered also the neoadjuvant setting, with FDA approval of nivolumab after the first data from CheckMate 816. In a scenario in which more and more ICIs will be available for early-stage NSCLC pts, new methods guiding pts’ selection apart from the usual PD-L1 expression would be desirable.

Treatment options are expanding, however, in a world in which new drugs are being approved faster and faster, often using surrogate endpoints, caution must be kept. In particular, most clinical trials in the neoadjuvant setting are currently using MPR/pCR or EFS as primary endpoints, while none of these are validated surrogate endpoints for OS in NSCLC pts. Furthermore, no trials are currently comparing new drugs in the neoadjuvant versus adjuvant setting, and only indirect comparisons between EFS and DFS can actually be made. Another open question is relative to the effective benefit of ICIs in early-stage EGFRm and ALKr pts, who are currently excluded from most of the relative clinical trials (and no benefit was evidenced in these subgroups for example in IMpower010).

All things considered, a lot of unsolved controversies are yet to be settled, with the ultimate goal of improving pts selection with a better molecular characterization, to widen treatment options, and enhance survival outcomes in potentially cured pts.

## 6. Conclusions

The scenario of adjuvant and neoadjuvant therapies in early-stage NSCLC pts is going through a revolution thanks to the introduction of targeted agents and immunotherapy. Several questions are still open in optimizing their therapeutic paths but the future seems bright. Ongoing clinical trials and mature data from completed ones will certainly help in casting a light on remaining doubts and controversies.

## Figures and Tables

**Figure 1 ijms-23-07222-f001:**
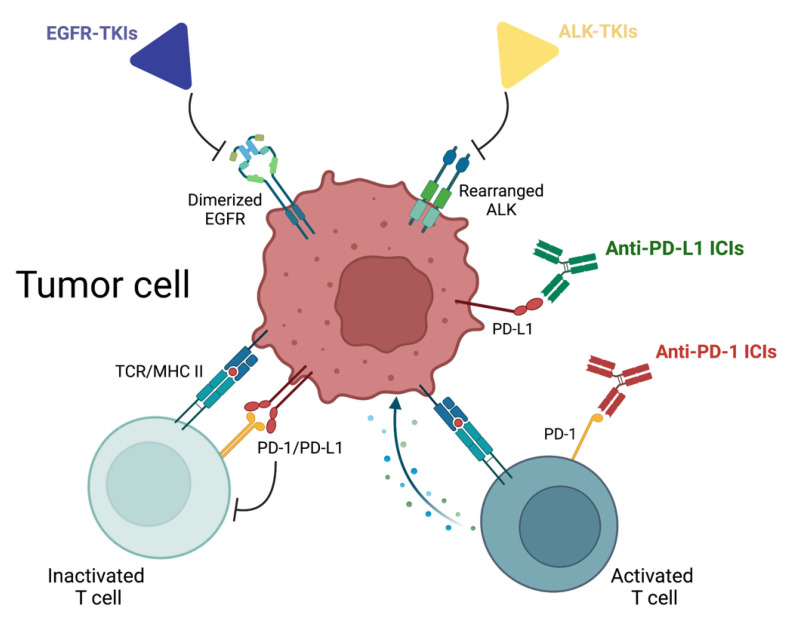
Mechanisms of action of main drugs evaluated in the adjuvant and neoadjuvant setting in NSCLC patients (created with BioRender.com (accessed on 15 June 2022)). *Abbreviations: EGFR, Epidermal Growth Factor Receptor; ALK, Anaplastic Lymphoma Kinase; TKIs, tyrosine kinase inhibitors; PD-(L)1, Programmed death-(ligand) 1; ICIs, immune checkpoint inhibitors; TCR, T-cell receptor; MHC II, major histocompatibility complex*.

**Figure 2 ijms-23-07222-f002:**
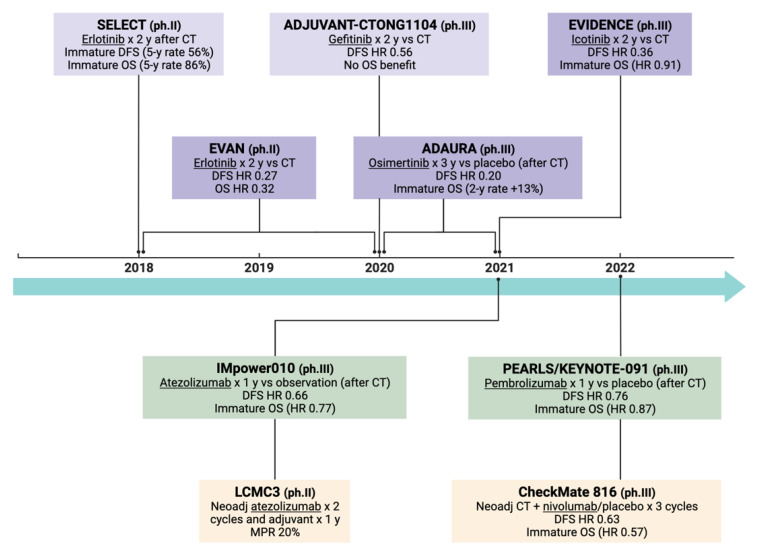
Main clinical trials in the adjuvant/neoadjuvant setting with EGFR-TKIs and ICIs with available literature data (created with BioRender.com (accessed on 15 June 2022)). *Abbreviations: y, years; CT, chemotherapy; DFS, disease-free survival; OS, overall survival; neoadj, neoadjuvant; MPR, major pathological response; EGFR-TKIs, Epidermal Growth Factor Receptor*
*tyrosine kinase inhibitors; ICIs, immune checkpoint inhibitors*.

**Table 1 ijms-23-07222-t001:** Clinical trials with available data with EGFR-TKIs in the adjuvant setting.

Clinical Trial	Phase	N° pts ^a^	Years	Stage	Treatment Arms	DFS	OS
**BR19** [10] (NCT00049543)	III	503 (EGFRm- unselected)	2002–2005	IB-IIIA	Gefitinib × 2 y vs. placebo (after adj CT) (1:1)	No difference (HR 1.22, 95% CI 0.93–1.61, *p* = 0.15)	No difference (HR 1.24, 95% CI 0.94–1.64, *p* = 0.14)
**ADJUVANT-CTONG1104** [11] (NCT01405079)	III	222	2011–2014	II-IIIA	Gefitinib × 2 y vs. adj CT (1:1)	30.8 vs. 19.8 m (HR 0.56, 95% CI 0.40–0.79, *p* = 0.001)	75.5 vs. 62.8 m (HR 0.92, 95% CI 0.62–1.36, *p* = 0.674)
**IMPACT** [12] (UMIN000006252)	III	234	2011–2015	II-III	Gefitinib × 2 y vs. adj CT (1:1)	35.9 vs. 25.1 m (HR 0.92, 95% CI 0.67–1.28, *p* = 0.63)	No difference (HR 1.03, 95% CI 0.65–1.65, *p* = 0.89)
**RADIANT** [13] (NCT00373425)	III	973 (‘EGFR- positive’)	2007–2010	IB-IIIA	Erlotinib × 2 y vs. placebo (after adj CT) (2:1)	50.5 vs. 48.2 m (HR 0.90, 95% CI 0.74–1.10, *p* = 0.324)	Not reached (HR 1.13, 95% CI 0.88–1.45, *p* = 0.335)
**SELECT** [14] (NCT00567359)	II	100	2008–2012	IA-IIIA	Erlotinib × 2 y (after adj CT)	Not reached (5-year DFS rate 56%)	Not reached (5-year OS rate 86%)
**EVAN** [15,16] (NCT01683175)	II	102	2012–2015	IIIA	Erlotinib × 2 y vs. adj CT (1:1)	42.4 vs. 21.0 m (HR 0.27, 95% CI 0.14–0.53, *p* < 0.0001)	84.2 vs. 61.1 m (HR 0.32, 95% CI 0.15–0.67)
**EVIDENCE** [17] (NCT02448797)	III	322	2015–2019	II-IIIA	Icotinib × 2 y vs. adj CT (1:1)	47.0 vs. 22.1 m (HR 0.36, 95% CI 0.24–0.55, *p* < 0.0001)	Not reached (HR 0.91, 95% CI 0.42–1.94)
**ADAURA** [18,19] (NCT02511106)	III	682	2015–2019	IB-IIIA	Osimertinib × 3 y vs. placebo (after adj CT or not) (1:1)	Not reached vs. 27.5 m (HR 0.20, 99% CI 0.14–0.30, *p* < 0.001) ^b^	Not reached (2-year OS rate 98% vs. 85%) ^b^

^a^ Patients where EGFRm were not specified. ^b^ Data regarding overall population. *Abbreviations: EGFR-TKIs, Epidermal Growth Factor Receptor tyrosine kinase inhibitors; N° pts, number of patients; DFS, disease-free survival; OS, overall survival; EGFRm, EGFR mutations; y, years; adj, adjuvant; CT, chemotherapy; m, months.*

**Table 2 ijms-23-07222-t002:** Ongoing clinical trials with EGFR-TKIs in the adjuvant setting.

Clinical Trial	Phase	N° pts	Estimated Primary Completion	Stage	Treatment Arms	Primary Endpoint
NCT02518802	III	220	Jan 2018	II-IIIA	Gefitinib × 2 y started during or after CT vs. adj CT	DFS
NCT03381430	II	50	Mar 2023	IIIA N2	Gefitinib × 2 y + adj RT	DFS
NCT02193282	III	450 ^a^	Oct 2026	IB-IIIA	Erlotinib × 2 y vs. placebo (after adj CT)	OS
**ICWIP** [20] (NCT02125240)	III	124	Dec 2018	II-IIIA	Icotinib × 3 y vs. placebo	DFS
**ICTAN** (NCT01996098)	III	318	Jan 2020	II-IIIA	Icotinib × 6 m vs. icotinib × 12 m vs. observation (after adj CT)	DFS
NCT03983811	III	174	Oct 2021	IIB-IIIA	Icotinib/placebo on days 8–15 during adj q21 CT cycles, then × 2 y	DFS
**CORIN** (NCT02264210)	II	128	Dec 2025	IB	Icotinib × 12 m vs. observation	DFS
NCT01746251	II	92	Nov 2020	I-III	Afatinib × 3 m vs. afatinib × 2 y	RFS
**ADAURA2** (NCT05120349)	III	380	Aug 2027	IA2-IA3	Osimertinib × 3 y vs. placebo	DFS
**FORWARD** (NCT04853342)	III	318	Dec 2023	II-IIIA	Furmonertinib vs. placebo (after adj CT)	DFS
**ATHEM** (NCT05165355)	II	90	Nov 2024	IB-IIA ^b^	Furmonertinib × 3 y	DFS
NCT04687241	III	192	Jan 2026	II-IIIB N2	Almonertinib vs. placebo (after adj CT)	DFS
**APEX** (NCT04762459)	III	606	May 2026	II-IIIA	Almonertinib × 3 y vs. almonertinib + adj CT vs. adj CT (3:2:1)	DFS

^a^ Trial arm with EGFRm patients. ^b^ Patients with high-risk pathological subtypes. *Abbreviations: EGFR-TKIs, Epidermal Growth Factor Receptor tyrosine kinase inhibitors; N° pts, number of patients; y, years; CT, chemotherapy; adj, adjuvant; DFS, disease-free survival; RT, radiotherapy; OS, overall survival; m, months; RFS, recurrence-free survival.*

**Table 3 ijms-23-07222-t003:** Ongoing clinical trials with EGFR-TKIs in the neoadjuvant (+adjuvant) setting.

Clinical Trial	Phase	N° pts	Estimated Primary Completion	Stage	Treatment Arms	Primary Endpoint
NCT03656393	III	48	Jul 2020	II-IIIA	Gefitinib × 56 d vs. CT × 6 w (+ adj CT if not responding disease)	2-year DFS rate
NCT03203590	III	590	Jan 2026	II-IIIA	Gefitinib × 8 w vs. CT × 2 cycles	2-year DFS rate
NCT03749213	II	36	Feb 2022	IIIA N2	Neoadj icotinib × 8 w, then × 2 y after surgery	ORR
**Neoafa** (NCT04470076)	II	30	Dec 2021	II-IIIB	Neoadj CT + afatinib (48 h after and until 24 h before CT) × 3 cycles, then adj afatinib × 2 y after surgery	MPR, ORR
NCT03433469	II	27	Dec 2022	I-IIIA	Neoadj osimertinib × 1–2 cycles	MPR
**NeoADAURA** [28] (NCT04351555)	III	328	Mar 2024	II-IIIB N2	Neoadj osimertinib + CT × 3 cycles vs. placebo + CT vs. osimertinib alone (1:1:1)	MPR

*Abbreviations: EGFR-TKIs, Epidermal Growth Factor Receptor tyrosine kinase inhibitors; N° pts, number of patients; d, days; w, weeks; CT, chemotherapy; adj, adjuvant; DFS, disease-free survival; y, years; ORR, objective response rate; neoadj, neoadjuvant; MPR, major pathological response.*

**Table 4 ijms-23-07222-t004:** Ongoing clinical trials with ALK-TKIs in the (neo-)adjuvant setting.

Clinical Trial	Phase	N° pts	Estimated Primary Completion	Stage	Treatment Arms	Primary Endpoint
**ALCHEMIST** [34] (NCT02194738)	III	8300 ^a^	Sep 2026	IB-IIIA	Crizotinib × 2 y vs. observation (after adj CT)	OS
**ALINA** [35] (NCT03456076)	III	257	Jun 2023	IB-IIIA	Alectinib × 2 y vs. adj CT	DFS
NCT05341583	III	202	Jun 2025	II-IIIB	Ensartinib × 2 y vs. placebo	DFS
NCT05186506	II	152	Dec 2025	II-IIIA	Ensartinib × 2 y vs. adj CT	DFS
NCT05241028	II	80	Feb 2027	IB-IIIA	Ensartinib × 3 y (after adj CT)	3-year DFS rate
**ALNEO** [36] (NCT05015010)	II	33	May 2023	III	Neoadj alectinib × 8 w, then adj × 96 w after surgery	MPR
**NAUTIKA1** (NCT04302025)	II	80 ^a^	Mar 2023	IB-III	Neoadj alectinib × 8 w, then adj CT and alectinib × 2 y	MPR

^a^ Population for the whole trial, independently from oncogenic drivers. *Abbreviations: ALK-TKIs, Anaplastic Lymphoma Kinase tyrosine kinase inhibitors; N° pts, number of patients; y, years; adj, adjuvant; CT, chemotherapy; OS, overall survival; DFS, disease-free survival; neoadj, neoadjuvant; w, weeks; MPR, major pathological response.*

**Table 5 ijms-23-07222-t005:** Clinical trials with available data with ICIs in the (neo-)adjuvant setting.

Clinical Trial	Phase	N° pts	Years	Stage	Treatment Arms	DFS	OS
**IMpower010** [37] (NCT02486718)	III	1280	2015–2018	IB-IIIA	Atezolizumab × 1 y vs. observation (after adj CT) (1:1)	Not reached vs. 35.3 m (HR 0.66, 95% CI 0.50–0.88, *p* = 0.0039) ^a^	Immature data (HR 0.77, 95% CI 0.51–1.17) ^a^
**PEARLS/KEYNOTE-091** [38] (NCT02504372)	III	1177	2015–2021	IB-IIIA	Pembrolizumab × 1 y vs. placebo (after adj CT) (1:1)	53.6 vs. 42.0 m (HR 0.76, 95% CI 0.63–0.91, *p* = 0.0014)	Immature data (HR 0.87, 95% CI 0.57–1.18, *p* = 0.14)
**LCMC3** [39] (NCT02927301)	II	181	2017–2020	IB-IIIB N2	Neoadj atezolizumab × 2 cycles, then adj atezolizumab × 1 y	Not available (primary endpoint MPR 20%)	Not available
**CheckMate 816** [40] (NCT02998528)	III	773	2017–2019	Ib-IIIA	Neoadj CT + nivolumab/placebo × 3 cycles (1:1)	31.6 vs. 20.8 m ^b^ (HR 0.63, 97% CI 0.43–0.91, *p* = 0.005)	Immature data (HR 0.57, 99% CI 0.30–1.07)

^a^ Data regarding stage II-IIIA population with PD-L1 ≥ 1% (primary endpoint). ^b^ Data on EFS. *Abbreviations: ICIs, immune checkpoint inhibitors; N° pts, number of patients; DFS, disease-free survival; OS, overall survival; y, years; adj, adjuvant; CT, chemotherapy; neoadj, neoadjuvant; MPR, major pathological response; EFS, event-free survival.*

**Table 6 ijms-23-07222-t006:** Ongoing clinical trials with ICIs in the (neo-)adjuvant setting.

Clinical Trial	Phase	N° pts	Estimated Primary Completion	Stage	Treatment Arms	Primary Endpoint
NCT04367311	II	100 ^a^ (ctDNA+)	Jan 2023	IB-IIIA	CT + atezo × 4 cycles, then atezo × 13 more cycles	% pts ctDNA- ^b^
**ACCIO** [41] (NCT04267848)	III	1210	Dec 2024	II-IIIB	CT + concomitant pembro × 4 cycles, then pembro × 13 cycles vs. CT + sequential pembro × 17 cycles vs. CT (1:1:1)	DFS
NCT04317534	II	368	Apr 2025	I	Pembro q42 × 9 cycles vs. observation (1:1)	DFS
**ANVIL** [42] (NCT02595944)	III	903	Jul 2024	IB-IIIA	Nivolumab × 1 y vs observation (after adj CT) (1:1)	DFS, OS
**NADIM-** **ADJUVANT** [43] (NCT04564157)	III	210	Apr 2027	IB-IIIA	CT + nivolumab q21 × 4 cycles, then nivolumab q28 × 6 cycles vs. CT × 4 cycles (1:1)	DFS
**BR31** (NCT02273375)	III	1415	Jan 2023	IB-IIIB N2	Durvalumab × 1 y vs observation (after adj CT) (1:1)	DFS ^c^
**MERMAID-1** [44] (NCT04385368)	III	332	Dec 2024	II-III	CT + durvalumab/placebo q21 × 4 cycles, then durvalumab/placebo q28 × 1 y (1:1)	DFS in ctDNA+
**MERMAID-2** [45] (NCT04642469)	III	284 ^a^ (ctDNA+)	Nov 2025	II-III	Durvalumab vs. placebo × 2 y (after (neo-)adj CT) (1:1)	DFS ^d^
**LungMate-008** (NCT04772287)	III	341	Dec 2026	II-IIIB N2	Toripalimab vs. placebo × 4 cycles (after adj CT)	DFS
**IMpower-030** [46] (NCT03456063)	III	451	Nov 2024	II-IIIB N2	Neoadj CT + atezo × 4 cycles, then adj atezo × 16 cycles vs. neoadj CT + placebo and no adj treatment	EFS
NCT04832854	II	82	Feb 2027	II-IIIB N2	Neoadj CT + atezo + tiragolumab × 4 cycles, then adj atezo + tiragolumab × 16 cycles	MPR
**KEYNOTE-671** [47] (NCT03425643)	III	786	Jan 2024	II-IIIB N2	Neoadj CT + pembro/placebo × 4 cycles, then adj pembro/placebo × 13 cycles (1:1)	EFS, OS
**INNWOP1** (NCT04875585)	II	33	Dec 2023	I-IIIA	Neoadj pembro + lenvatinib × 6 w, then adj pembro × 15 cycles	MPR
**CANOPY-N** (NCT03968419)	II	88	Apr 2022	IB-IIIA (no T4 or N2)	Neoadj pembrolizumab vs. canakinumab vs. pembrolizumab + canakinumab × 2 cycles	MPR
**CheckMate 77T** [48] (NCT04025879)	III	452	Dec 2023	II-IIIB N2	Neoadj CT + nivolumab/placebo, then adj nivolumab/placebo	EFS
**GECP 16/03_NADIM** (NCT03081689)	II	46	Jun 2022	IIIA N2	Neoadj CT + nivolumab × 3 cycles, then adj nivolumab × 1 y	2-year PFS rate
**NADIM II** (NCT03838159)	II	86	Nov 2026	IIIA-IIIB N2	Neoadj CT + nivolumab/placebo, then adj nivolumab/observation	pCR
**NeoCOAST 2** (NCT05061550)	II	140	Feb 2026	II-IIIA	Neoadj CT + durvalumab + monalizumab/oleclumab q21 × 4 cycles, then adj monalizumab/oleclumab q28 (1:1)	pCR rate
**AEGEAN** [49] (NCT03800134)	III	824	Apr 2024	II-IIIB N2	Neoadj CT + durvalumab/placebo q21 × 4 cycles, then adj durvalumab/placebo q28 × 12 cycles (1:1)	pCR rate, EFS

^a^ Patients with detectable ctDNA after surgery. ^b^ Percentage of patients with undetectable ctDNA after adjuvant treatment. ^c^ DFS in patients with PD-L1 ≥ 25%. ^d^ DFS in patients with PD-L1 ≥ 1%. *Abbreviations: ICIs, immune checkpoint inhibitors; N° pts, number of patients; ctDNA, circulating tumor DNA; CT, chemotherapy; atezo, atezolizumab; DFS, disease-free survival; pembro, pembrolizumab; y, years; adj, adjuvant; neoadj, neoadjuvant; EFS, event-free survival; MPR, major pathological response; w, weeks; pCR, pathologic complete response.*

## Data Availability

Not applicable.

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
