# Peer review of "Targeted Therapy and Immunotherapy in Early-Stage Non-Small Cell Lung Cancer: Current Evidence and Ongoing Trials"

_ijms, 2022, doi:10.3390/ijms23137222_

Round 1

Reviewer 1 Report

Targeted tumor therapies and immunotherapies are more and more important in the treatment of cancer.  In their review, the authors summarized completed and ongoing trials with targeted agents and immunotherapy in early-stage non-small cell lung cancer with focus on practice-changing results and new perspectives for potentially curable patients.  The authors presented the study design and, if available, results of 53 phase II and III clinical trials.  The presentation of the data is careful and accurate and summarized in six tables for epidermal growth factor receptor tyrosine kinase inhibitors, anaplastic lymphoma kinase tyrosine kinase inhibitors, and immune checkpoint inhibitors.  The general structure of the review is consistent and appropriate and provides an important comparison of different treatment options and their outcome.  The cited literature is up to date.  However, the way the text is presented is very tiring to read and the well-presented tables cannot compensate for this.  The entire presentation is too descriptive and not very evaluative, so that a reader who is not of the specific subject cannot draw sufficient conclusions from the data.  Moreover, data from the European Union Clinical Trials Register are missing.

Specific comments:

1.     It must be possible to read the abstract as a stand-alone work.  Therefore, all abbreviations must already be defined in the abstract, which is not the case.  It is not recommended to abbreviate “patients”.

2.     FDA and EMA are not defined at all.

3.     The authors only refer (with one exception) to the FDA and all listed clinical trials (with one exception, UMIN000006252) are from the U. S. National Library of Medicine.  The authors should also include, though there is some overlap, the European Union Clinical Trials Register and should not only refer to the FDA but also to the EMA, which also plays an important role.

4.     The authors should also provide figures.  At least two figures seems to be of high benefit for the readers, (1) a graphical representation illustrating the points of attack of the respective treatment methods, and (2) a chart that graphically summarizes the most important results shown in the tables, but not only as a repetition of the results presented in the tables but in some kind of order that allows the reader who is new in the field to see at a glance which treatment options appear to have what increase in efficacy, benefit or harm over others.

5.     The review is too descriptive, and a real discussion is missing.  A reader who is new in the field must be better guided through the different aspects.  The interpretation of the presented data should be discussed in more detail in a section “Discussion” addressing advantages, disadvantages, contradictory results or interpretations, deficiencies, missing data, specific comparisons, and an own assessment and evaluation of the data.  This can include aspects already presented in the “Conclusions” section.  The “Conclusions” itself should then be rather short, typically one paragraph, with a final summarizing statement and a short outlook.

Reviewer 2 Report

Line 33; TNM: please provide full name followed by abbreviation in parenthesis.

Reviewer 3 Report

This is an interesting and well-written review article describing the targeted therapy and immunotherapy in early-stage NSCLC. The manuscript reveals that the clinical trials published until recently and the ongoing studies were well summarized and described. In addition, the limitation of the current studies and future research directions are appropriately described. It's worth accepting without any revision.

Round 2

Reviewer 1 Report

The revised version of the manuscript is substantially improved and in particular the discussion is now of high value for the reader.